# Position: LLMs Need a Bayesian Meta-Reasoning Framework for More Robust and Generalizable Reasoning

Hanqi Yan [* 1]   Linhai Zhang [1]   Jiazheng Li [1]   Zhenyi Shen [1]   Yulan He [* 1 2]

## Abstract

Large language models (LLMs) excel in many reasoning tasks but continue to face significant challenges, such as lack of robustness in reasoning, struggling with cross-task generalization, and inefficiencies in scaling up reasoning capabilities. Current training paradigms, including next-token prediction and reinforcement learning from human feedback, often fall short in adaptability to diverse reasoning tasks. Existing approaches, such as prompt optimization and iterative output refinement, offer performance improvement, but can be inefficient and lack effective generalization. To overcome these limitations, this position paper argues for a transformative shift in how LLMs approach reasoning. Drawing inspiration from cognitive science, particularly meta-reasoning theories such as Dual-Process Theory and Metacognitive Reasoning, we propose a Bayesian meta-reasoning framework for LLMs. Our approach integrates self-awareness, monitoring, evaluation, regulation, and meta-reflection, to enhance LLMs' ability to refine reasoning strategies and generalize across tasks. We revisit existing LLM reasoning methods, identify key challenges, and suggest directions for future research. We provide a repository[1] with the resources referenced in our paper.

## 1. Introduction

Large language models (LLMs) have demonstrated remarkable potential in various reasoning tasks (Wei et al., 2022a; Hao et al., 2023a; Shao et al., 2024). Despite these advancements, they still face significant limitations, including the generation of hallucination with high confidence (Singh et al., 2023; Wen et al., 2024), vulnerability in trivial input perturbations (Wu et al., 2024a), and a lack of cross-task

---

[1]Department of Informatics, King's College London [2]The Alan Turing Institute. Correspondence to: Hanqi Yan <hanqi.yan@kcl.ac.uk>, Yulan He <yulan.he@kcl.ac.uk>.

*Proceedings of the $42^{nd}$ International Conference on Machine Learning*, Vancouver, Canada. PMLR 267, 2025. Copyright 2025 by the author(s).

[1]https://github.com/hanqi-qi/LLM_MetaReasoning

generalizability (Wu et al., 2024d; Qin et al., 2023).

To understand these limitations, we revisit the prevalent approaches for LLM reasoning. Most LLMs are built on the GPT backbone (Brown et al., 2020), which employs the *statistical next-token prediction* **pretraining** paradigm. As argued by McCoy et al. (2023), the sensitivity of LLMs to task frequency, input perturbations, and output tendencies can be traced back to their reliance on language frequency patterns. Secondly, *reinforcement learning (RL) algorithm (Ouyang et al., 2022; Shao et al., 2024)* with reward models are applied during reasoning-specific **post-training**. For verifiable tasks with clear answers, such as math and coding, the stepwise or outcome-level accuracy rewards are commonly used in the state-of-the-art models, such as OpenAI-o1 and DeepSeek-R1. For free-form questions without fixed ground truth, reward models trained on task-specific preference data are often used to provide feedback (DeepSeek-AI et al., 2024). However, these approaches depend on task-specific annotations, which limit scalability and generalizability in tasks where preference annotations are hard to obtain, such as causal reasoning (Chi et al., 2024) and Scientific discovery (Bazgir et al., 2025; Xiang et al., 2025b). Beyond the trained reward models, *LLM-generated feedback* is also used to iteratively refine model outputs (Xie et al., 2023; Shinn et al., 2023b). However, studies show that self-generated feedback is often unreliable (Yan et al., 2024b; Chen et al., 2025a), and sample-wise feedback fails to capture shared patterns underlying multiple cases (Yang et al., 2024; 2025a).

These limitations stem from LLMs being trained to solve tasks individually, rather than to learn *how* to arrive at those solutions. Ideally, they should develop the ability to recombine fundamental reasoning skills when faced with novel problems, thereby achieving a better generalization to unseen tasks. To achieve this, we need to move beyond the existing autoregressive reasoning system. Instead, **new learning paradigms** are needed—ones that empower models to actively engage in **learning-to-reason** or **meta-reasoning processes**. It envisions reasoning as an **adaptive process in which models not only solve tasks but also learn to improve their reasoning strategies over time.**

Emergent LLM meta-reasoning approaches largely rely on LLM prompting, such as comparing multiple thought pro-

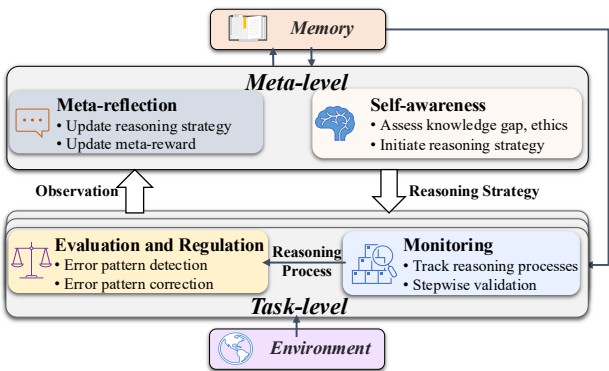

Figure 1. Our proposed Bayesian Meta-Reasoning framework.

cesses and identifying the most relevant information to answer questions (Yoran et al., 2023), or storing high-level "thought-templates", derived from past problem-solving processes, to guide future tasks (Yang et al., 2024; Gao et al., 2024a; Xiang et al., 2025a). However, none of them explores new learning or inference paradigms specifically for meta-reasoning, and most focus narrowly on math tasks.

In cognitive science, meta-reasoning theories explain how individuals monitor and regulate their reasoning processes. Dual-Process Theory (Kahneman, 2011) suggested that reasoning involves both intuitive and deliberative systems, with meta-reasoning balancing the two. Koriat (2000) highlighted the "*Feeling of Knowing*", which suggests that implicit and automatic feedback is actively involved in controlling the reasoning process. "*Feeling of Errors*" (Gangemi et al., 2015) explored how individuals detect errors and adjust their reasoning accordingly.

Inspired by the meta-reasoning research in cognitive science and building on advancements in Bayesian models for LLM reasoning (Xie et al., 2021; McCoy et al., 2023; Zhi-Xuan et al., 2024; Feng et al., 2025b), we propose a cognitive architecture for LLM meta-reasoning in Figure 1. It comprises several components in either the meta-level or task-level, each of which could be an LLM agent or an external module. Before generating reasoning steps for the given question, *Self-awareness* firstly analyzes the task via reviewing its own knowledge and initializing a reasoning strategy. Given the strategy, *Monitoring* tracks and evaluates stepwise reasoning using an overarching reward [2] beyond sample-wise annotation. *Evaluation and Regulation* reviews the overall reasoning process, detects common errors across multiple samples, and makes corrections. *Meta-reflection* explores alternative reasoning strategies and refines the meta-reward in *Memory*. LLM meta-reasoning may also combine with *External solvers* (in *Environment*), such as logic engines and calculators, which complement LLMs through verifiable outputs grounded in rigorous methodologies. This process iterates and aims to improve the LLMs' reasoning quality.

---

[2] Noted that the term *reward* in our paper broadly refers to general feedback, not limited to verified reward.

**Paper Structure.** Section 2 discusses open problems in LLM reasoning, motivating our Bayesian Meta-reasoning Framework in Section 3. Section 4 analyzes gaps in existing reasoning approaches. Section 5 outlines future directions. Finally, Section 6 presents alternative views and Section 7 concludes the paper.

## 2. Arguments for Meta-Reasoning in LLMs

This section outlines LLM open problems and highlights the potential of the meta-reasoning paradigm to address them.

**Open Problem 1.** *LLMs often exhibit a strong "Feeling of Knowing" but lack crucial human-like cognitive attributes, such as "awareness of limitations"(Gangemi et al., 2015) and "awareness of situation" (Zhan et al., 2024).*

LLMs should develop self-awareness to critically evaluate whether a given task aligns with their knowledge and reasoning capabilities before proceeding. This capability would help mitigate hallucinations, discourage attempts at solving unsolvable problems, and prevent engagement in unethical tasks, ensuring more responsible and reliable behavior.

**Open Problem 2.** *LLMs lack the adaptivity to incorporate question-tailored strategies, which can lead to inefficiency and reduced generalizability across tasks (Sprague et al., 2025; Liu et al., 2024d). For instance, Liu et al. (2024d) identified cognitive tasks where deliberation hinders human performance and observed similar limitations in LLMs when using Chain-of-Thought (CoT) reasoning.*

Before tackling a problem, LLMs should formulate an abstract strategy based on the problem's structure, rather than relying on superficial cues like entities or phrasing. For example, counterfactual thinking can be applied to infer causality across diverse scenarios. Furthermore, through reflective processes, LLMs should be able to dynamically refine their reasoning strategies, such as incorporating temporal coherence in counterfactual thinking. This refinement involves learning from errors across multiple instances, ultimately benefiting overall task performance.

**Open Problem 3.** *LLMs struggle with complex planning and generalizable reasoning. RL with predefined reward often overfits to simplistic reward structures, leading to reward hacking (Skalse et al., 2022; Eisenstein et al., 2024; Qin et al., 2024), where agents exploit flaws in the reward function to achieve high scores without genuinely learning transferable reasoning patterns.*

For human, the development of problem-solving skills does not stem from learning isolated facts across separate cases, but from longitudinal adaptation (Flavell, 1979). For LLMs, this implies moving beyond alignment with case-wise annotated reasoning steps, toward training objectives that evolve over time and generalize across examples, such as enhancing efficiency or achieving balanced learning across tasks.

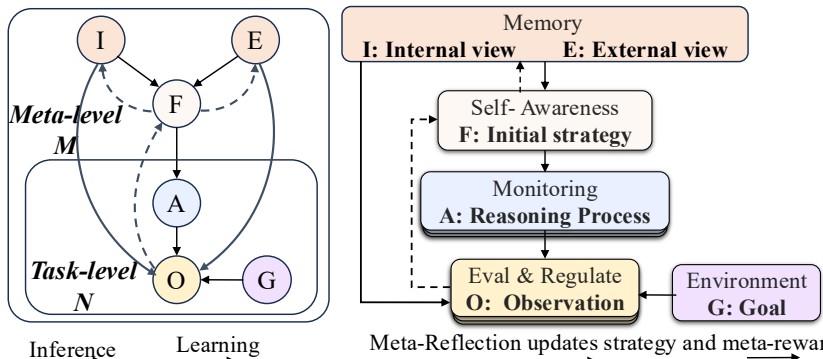

*Figure 2.* **Left:** The Bayesian framework with both task-level and meta-level components. **Right:** The definitions of variables and their associated modules in the meta-reasoning framework.

**Open Problem 4.** *LLMs struggle to efficiently internalize new knowledge. Current methods, such as on-the-fly knowledge retrieval or model fine-tuning, fail to adequately address the knowledge conflicts and resource inefficiency.*

Humans do not overhaul their entire cognitive framework when learning new skills; instead, they selectively refine and build upon prior knowledge. Similarly, LLMs require modular and targeted updates to avoid catastrophic forgetting while maintaining resource efficiency.

## 3. Conceptualized Framework

To achieve the aforementioned characteristics, we propose a Bayesian framework shown in Figure 2.

### 3.1. Bayesian Inference and Learning Processes

**Bayesian Inference.** In our framework, we conceptualize the latent variables as $\mathbf{I}$, $\mathbf{E}$ (can be combined as $\Theta = \{\Theta_I, \Theta_E\}$) and $\mathbf{F}$, while the generated reasoning process is $\mathbf{A}$ and the observation is $\mathbf{O}$. The bi-level inference is formalized as follows;

$$\text{Meta-level:} \quad p(\mathbf{F}|\Theta_\mathbf{I}, \Theta_\mathbf{E}),$$
$$\text{Task-level:} \quad p(\mathbf{O}|\mathbf{A}), p(\mathbf{A}|\mathbf{F}),$$

The joint posterior is:

$$p(\Theta_\mathbf{I}, \Theta_\mathbf{E}, \mathbf{F} \mid \mathbf{O}) = \frac{p(\mathbf{O} \mid \mathbf{I}, \mathbf{E}, \mathbf{F})p(\mathbf{I}, \mathbf{E}, \mathbf{F})}{p(\mathbf{O})}, \quad (1)$$

where:

$\Theta_I$ (*Internal View*) represents inherent and foundational knowledge, akin to long-term memory, such as world knowledge encoded during pre-training.

$\Theta_E$ (*External View*) is task-specific knowledge, comparable to working memory, which dynamically updates based on input and context. Examples include temporarily retrieved facts or intermediate reasoning steps generated.

$\mathbf{F}$ (*Reasoning Strategy*) is essential to the *learning-to-reason* framework, as it represents an adaptive strategy tailored to

the skills required for different input tasks.

$\mathbf{O}$ (*Observation*) is the evaluated results of $\mathbf{A}$, based on both the environment $\mathbf{G}$, e.g., the sampled trajectories from an external reference model, and internal mechanisms, e.g., generation confidence.

**Bayesian Learning.** It is employed to optimize model parameters through bi-level updates for both the reasoning strategy $\mathbf{F}$ at the task-level, and the knowledge priors $\Theta_I$ and $\Theta_E$ at the meta-level. This process ensures alignment with both task-specific and overarching rewards.

*Updating reasoning strategies at the task-level.* The goal is to update the posterior distribution of reasoning strategies $\mathbf{F}$ given the observations $\mathbf{O}$, foundational knowledge $\Theta_I$, and task-specific knowledge $\Theta_E$:

$$p(\mathbf{F}|\mathbf{O}, \Theta_I, \Theta_E) \propto p(\mathbf{O}|\mathbf{F})p(\mathbf{F}|\Theta_I, \Theta_E), \quad (2)$$

where $p(\mathbf{F}|\Theta_I, \Theta_E)$ is the prior of reasoning strategies, informed by the model's internal knowledge ($\mathbf{I}$) and task-specific knowledge ($\mathbf{E}$), and $p(\mathbf{O}|\mathbf{F}, \Theta_I, \Theta_E)$ is the likelihood of the observation given the reasoning strategy and the priors. The method of updating $\mathbf{F}$ using observation and rewards is shown in Table 1. This iteration continues until it identifies a $\mathbf{F}$ that maximizes the cumulative reward. The implementation of how to derive the cumulative reward $\mathcal{R}$ can be found in Section 4.4: inverse planning.

*Table 1.* Algorithm of updating reasoning strategy.

| |
| --- |
| 1. Choose an initial reasoning strategy $\mathbf{F} \sim p(\mathbf{F}|\Theta_I, \Theta_E)$ |
| 2. Generate reasoning process $\mathbf{A} = \{(\mathbf{s}_i, \mathbf{a}_i)\}_{i=1}^T$ using the current $\mathbf{F}$. |
| 3. Compute feedback $\mathbf{O}$ using $\mathbf{A}$ and the reward $\mathcal{R}(\mathbf{F}; \mathbf{O}, \Theta_I, \Theta_E)$. |
| 4. Update $\mathbf{F}$ with Eq. (2), where $p(\mathbf{O}|\mathbf{F}) \propto \exp(\mathcal{R}(\mathbf{F}, \Theta_I, \Theta_E))$. |

*Updating knowledge priors at the meta-level.* $\Theta_\mathbf{I}$ (foundational knowledge) and $\Theta_\mathbf{E}$ (task-specific knowledge) act as priors that shape the reasoning strategy $\mathbf{F}$. They can also be refined based on performance feedback and meta-rewards. This ensures that reasoning strategies not only improve for a given task but also generalize effectively across different tasks and domains. The posterior over knowledge priors is:

$$p(\Theta_\mathbf{I}, \Theta_\mathbf{E}|\mathbf{O}, \mathbf{F}) \propto \int p(\mathbf{O}|\mathbf{F})p(\mathbf{F}|\Theta_\mathbf{I}, \Theta_\mathbf{E}) \, d_\mathbf{F} p(\Theta_\mathbf{I})p(\Theta_\mathbf{E}),$$

where $p(\mathbf{O}|\mathbf{F})$ is the likelihood of the observed reasoning process. For each reasoning episode, we can collect observations $\mathbf{O}$ and cumulative meta-rewards $\mathcal{R}$. $\mathcal{R}$ could evaluate both $\mathbf{I}$ and $\mathbf{E}$ given the updated reasoning strategy. For example, if a reasoning strategy $\mathbf{F}$ derived from $\Theta_{\mathbf{I}}$ and $\Theta_{\mathbf{E}}$ succeeds across multiple tasks, then knowledge priors should be strengthened, otherwise, if knowledge priors lead to contradictions or inefficiencies, they should be adjusted.

## 3.2. Core Components of the Framework

This framework integrates foundational knowledge, task-specific adaptation, and reasoning strategies to enable dynamic and adaptive reasoning processes.

***Memory*** stores the two priors, $p(\Theta_{\mathbf{I}})$ and $p(\Theta_{\mathbf{E}})$, representing inherent and task-specific insights essential for developing adaptive reasoning strategies and rewards.

***Self-awareness*** evaluates the skill gap between the input task and the model's capabilities. Based on this evaluation, it proposes an initial reasoning strategy with latent skill distribution, $p(\mathbf{F}|\Theta_{\mathbf{I}}, \Theta_{\mathbf{E}})$. The reasoning strategy is not a direct solution to the task but serves as a *meta-level planner*. For example, CoT is beneficial for tasks requiring multi-step logical deduction (e.g., algorithmic problem-solving). In contrast, direct answers are more efficient for tasks that rely on simple knowledge recall, where excessive deliberation can hinder performance and computational efficiency (Sabbata et al., 2024; Liu et al., 2024d).

***Monitoring*** performs *stepwise* validation based on the reasoning strategy $\mathbf{F}$. Without loss of generalizability, we represent a sequence of $T$ state-action pairs as $\mathbf{A} = [(\mathbf{s}_0, \mathbf{a}_0), (\mathbf{s}_1, \mathbf{a}_1), \ldots, (\mathbf{s}_{T-1}, \mathbf{a}_{T-1}), \mathbf{s}_T]$, which can be evaluated by a reward model $\mathbf{Q}_t$. The reward model leverages cognitive resources like $\mathbf{I}$ and $\mathbf{E}$, focusing on task-agnostic criteria like contradictions (Zhang et al., 2024b) and task-specific proxy signals from task-related solutions.

***Evaluation and regulation*** are applied for evaluating and correcting the *final* reasoning process $\mathbf{A}$ using both internal resources and external solvers (e.g., a calculator generating grounding information $\mathbf{G}$ for arithmetic validation). Feedback $\mathbf{O}$ enables the framework to iteratively revise the reasoning process until the task requirements are satisfied.

***Meta-reflection*** integrates feedback ($\mathbf{O}$) from the environment or the system's own evaluation processes to iteratively improve its internal representations and strategies.

**Example of scientific hypothesis generation.** Here *Self-Awareness* corresponds to recognizing skill gaps when generating hypotheses. *Monitoring* tracks reasoning steps to ensure logical consistency and alignment with evidence. *Evaluation* reviews and compares hypotheses for soundness, feasibility, and relevance. *Regulation* modified the reasoning process to address inconsistencies or flawed assumptions.

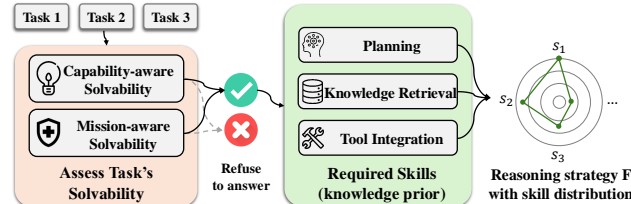

*Figure 3.* **Self-awareness**. It first assesses the task's solvability under capability awareness and mission awareness. For solvable tasks, it initializes the reasoning strategies based on knowledge priors. For the scientific hypothesis generation, the reasoning strategy is a distribution over multiple skills, such as *cross-domain analogy* (identify analogous phenomena across fields), *constraint satisfaction* (formulate hypotheses that meet known constraints), and *anomaly-driven exploration* (explain data anomalies).

*Meta-Reflection* examines the process for hypotheses generation and adapts reasoning strategies if necessary. *Memory* stores previous reasoning processes for iterative learning and generalization. We can also incorporate *External Solvers*, such as simulation models, to complement and verify LLM-generated reasoning steps. Our framework could enable more robust scientific reasoning.

## 4. Gaps and Limitations

We discuss reasoning approaches that share certain characteristics with our framework and highlight remaining gaps. A detailed review is in the Appendix.

### 4.1. Self-Awareness

Our self-awareness module (Fig. 3) comprises two core components, inspired by Ackerman & Thompson (2017): (1) *assessing the task's solvability* $p(\Theta_{\mathbf{I}})$ and $p(\Theta_{\mathbf{E}})$, based on the model's technic and cognitive capabilities, where capability-aware solvability and mission-aware solvability are considered simultaneously (2) *initializing reasoning strategies* $p(\mathbf{F}|\Theta_{\mathbf{I}}, \Theta_{\mathbf{E}})$ by analyzing the skill gaps between LLM itself and the task, and adaptively generating the most appropriate reasoning strategy to bridge the gaps and approach the task more effectively.

#### 4.1.1. ASSESSING THE TASK SOLVABILITY

The self-awareness of LLMs, treating LLMs as cognitive entities, has emerged as a frontier research area (Huang et al., 2024d). Li et al. (2024b) categorized self-cognition into five dimensions, i.e., *capability*, *mission*, *emotion*, *culture*, and *perspective*. We focus on the first two dimensions here. *Capability awareness* is crucial, as highlighted by the Dunning-Kruger Effect (Kruger & Dunning, 2000), a cognitive bias where individuals tend to overestimate their knowledge or abilities in a particular domain. *Mission awareness* evaluates whether LLMs understand their roles as AI models

designed to serve humanity, avoiding harmful or unethical actions (Huang et al., 2024c).

*Capability-aware Solvability.* While assessing capability-aware solvability of LLMs is underexplored, confidence or uncertainty estimation (Geng et al., 2024; Wen et al., 2024) offers a viable alternative. Applied to our setting, a confidence threshold can be set, below which the LLM outputs should be considered unreliable (Feng et al., 2024). Confidence and uncertainty estimation methods for LLMs can be categorized into white-box and black-box. *White-box methods* allow uncertainty estimation through token-level entropy (Huang et al., 2025) or utilizing attention weights or hidden states to build probing models (Kadavath et al., 2022; Burns et al., 2023; Azaria & Mitchell, 2023). *Black-box methods* rely solely on input-output behavior, without accessing to model internal parameters. Examples include prompting LLMs to express uncertainty verbally (Tian et al., 2023; Mielke et al., 2022) or inferring uncertainty by analyzing response agreement (Manakul et al., 2023).

*Mission-aware Solvability.* Although LLMs trained with RLHF (Ouyang et al., 2022) to align with human preferences, they may still produce harmful responses when presented with deliberatedly unethical requests (Shen et al., 2024; Deng et al., 2023). Some studies rely on small neural network models such as HateBERT (Caselli et al., 2021) and Perspective API (Lees et al., 2022) as the off-the-shelf toxicity detectors. Jailbreak defense aims to filter and reject malicious prompts (Xiong et al., 2024a; Mo et al., 2024; Liu et al., 2024a). Key approaches include fine-tuning models to reject harmful instructions (Mo et al., 2024), and adversarial training (Liu et al., 2024a), which exposes models to diverse attack scenarios (Liu et al., 2024a) to improve robustness.

**Limitation 1: *lack of a multi-view framework for task solvability.*** While existing studies have explored methods to measure an LLM's capability to solve a task, they lack a multi-view framework that integrates different perspectives on task solvability. Current approaches often rely on isolated measures, such as uncertainty estimation, but fail to provide a holistic decision-making process that considers efficiency, safety, and task relevance simultaneously. Moreover, balancing safety and utility remains challenging, as models may either take unnecessary risks or overly restrict themselves. A more comprehensive, multi-view approach is needed for LLM task solvability assessment. Actionable inisghts are in Section 5: Action 2.

### 4.1.2. Initializing the Reasoning Strategy

For solvable and ethical tasks, the next step is to propose a meta-level reasoning strategy, formulated before task execution. This strategy, modeled as a distribution over multiple skills, bridges the gap between the LLM's abilities and the task's requirements. Below, we detail this process by exploring three representative categories of skills: planning, external knowledge seeking, and tool execution.

*Planning Skills.* For tasks requiring step-by-step deduction, like multi-hop commonsense reasoning, CoT improves performance than direct prompting (Wei et al., 2022b). Tree of Thoughts (ToT) focuses on exploration skills, enabling the model to explore multiple parallel solution paths (Yao et al., 2023a), while Graph of Thought (GoT) emphasizes relational reasoning, making it suitable for tasks like knowledge graph navigation (Besta et al., 2024).

*Knowledge Retrieval Skills.* For tasks requiring up-to-date knowledge, such as Question Answering (Wang et al., 2024b) or fact-checking (Tang et al., 2024), knowledge retrieval bridges the skills gap. Adaptive Retrieval Augmented Generation (RAG) methods use probing datasets to determine when retrieval is necessary (Wang et al., 2023b). SELF-RAG integrates on-demand retrieval and self-reflection for generation quality (Asai et al., 2024).

*Tool Integration Skills.* For tasks beyond text processing, e.g., online shopping assistants (Yao et al., 2022) and code generation (Wang et al., 2024e), tool execution skills is needed. ChatCoT (Chen et al., 2023) integrates *Calculator* for math reasoning. ToolkenGPT (Hao et al., 2023b) offeres the flexibility to plug in an arbitrary number of tools by expanding the '*toolkens*'.

**Limitation 2: *lack of adaptability in latent skill selection.*** The methods reviewed above typically propose a single "optimal" strategy, often focusing on a specific skill dimension, such as planning. Ideally, as shown in Figure 3, a more flexible approach is needed, where a combination of latent skills is considered. The optimal reasoning strategy should vary not only across tasks but also across different instances of the same task, adaptively combining multiple skills tailored to each particular input. Actionable insights are in Section 5: Action 3.

### 4.2. Monitoring

Given a reasoning strategy $\mathbf{F}$ from the self-awareness module, *Monitoring* is employed to assess and control the generation of the reasoning process, guided by a reward model (shown in Figure 4). In LLM reasoning, we first sample $k$ candidate solutions at the $t$-th reasoning step, such as different intermediate steps for an arithmetic problem. These intermediate steps are then assessed by a reward model $Q_t$. Finally, a reasoning step $\mathbf{a}_t \in \mathbf{A}$ is sampled from the optimized policy model $\pi(\mathbf{a}_t|\mathbf{s}_t)$ with respective to the $Q_t$:

$$\mathbf{a}_t \sim \pi(\mathbf{a}_t \mid \mathbf{s}_t) = \frac{\exp(Q_t(\mathbf{s}_t, \mathbf{a}_t; \mathbf{F}; \Theta_{\mathbf{I}}, \Theta_{\mathbf{E}}))}{\sum_{\mathbf{a}' \in \mathcal{A}} \exp(Q_t(\mathbf{s}_t, \mathbf{a}'_t; \mathbf{F}; \Theta_{\mathbf{I}}, \Theta_{\mathbf{E}}))},$$

where the design of the reward model $Q_t$ is critical for stepwise reasoning assessment. We also need to design an iterative control mechanism for the entire reasoning process.

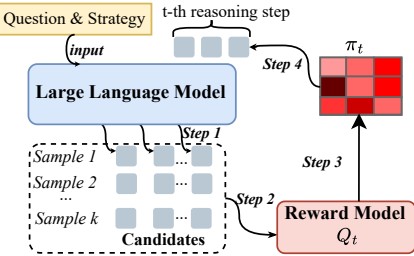

*Figure 4.* The **Monitoring** module assesses and controls the $t$-th reasoning step generation based on the policy model $\pi_t$ derived from the reward model $Q_t$. The reward model will provide overarching meta-reward beyond suboptimal heuristics.

### 4.2.1. REWARDS IN REASONING

Existing studies adopt either Outcome Reward Model (*ORM*)(Ouyang et al., 2022) or Process Reward Model (*PRM*)(Lightman et al., 2024; Wang et al., 2024c). *ORM* assesses the quality of the completed reasoning paths, while *PRM* provides fine-grained, even stepwise validation, thus mostly demonstrating superior performance in both model training and inference (Wang et al., 2024c; Lightman et al., 2024). These reward models can be derived by **training on human annotations** on the reasoning trajectories, such as classification (Lightman et al., 2024; Wang et al., 2024c), regression (Chen et al., 2024a; Wan et al., 2024) and pairwise preference (Ouyang et al., 2022; Xie et al., 2024); Or **advanced LLMs can be directly prompted** to provide feedback(Lightman et al., 2024; Lu et al., 2023). However, recent studies (Huang et al., 2024b; Li et al., 2024c; Yan et al., 2024b) reveal that LLMs often struggle to provide reliable feedback due to limited knowledge grounding. In contrast, Deepseek-R1 (DeepSeek-AI et al., 2025) combines two straightforward, **rule-based rewards**—a correctness reward on the final result and a format reward for adhering to the required response structure.

**Limitation 3:** *existing reward signals are imperfect proxy.*
Reward signals, apart from rule-based ones, include LLM-generated self-evaluations and task-specific reward models. Self-evaluations are often unreliable, displaying biases toward length, polished tone (Zeng et al., 2024b), and self-enhancement (Gu et al., 2024). Trained reward models, on the other hand, frequently rely on *oversimplified* objectives like correctness, format, failing to account for the multi-dimensional criteria (Wang et al., 2024a). Furthermore, these models are typically *stationary*, making them unsuitable for dynamic environments such as shifting data distributions during policy optimization (Gao et al., 2022). While methods like reward ensembles (Coste et al., 2024) and diverse feedback (Yu et al., 2023a) show promise, creating a robust model that evaluates intermediate reasoning steps and generalizes across scenarios remains an open challenge (Eisenstein et al., 2024). Actionable insights are in Section 5: Action 4.

### 4.2.2. POST-TRAINING WITH REWARDS

To post-train LLMs for better reasoning capabilities, we can leverage either Direct Preference Optimization (DPO) (Rafailov et al., 2023) or Reinforcement Learning with Human Feedback (RLHF). DPO focuses on training the model to better distinguish between desirable and undesirable trajectories in a contrastive manner, while RLHF relies on reward models to provide feedback to reasoning LLMs. **Rejection Sampling** (Dong et al., 2023) finetunes the model only on high-reward samples, aiming to shift the output distribution towards higher-quality samples, but it fails to leverage information from rejected samples. In contrast, **Preference Learning** (Grattafiori et al., 2024; Li et al., 2024a) uses all samples to train the model on preference pairs, either at the outcome level or at the process level. Initially implemented within a traditional RL framework, specifically Proximal Policy Optimization (PPO) (Schulman et al., 2017) with a trained reward model, preference learning has largely evolved toward DPO (Rafailov et al., 2023) due to its effectiveness and simplicity. More recently, **RL with Verifiable Rewards**, exemplified by models like Deepseek-R1 (DeepSeek-AI et al., 2025), trains policy models using Group Relative Policy Optimization (GRPO) (Shao et al., 2024), which removes the need for a critic model by estimating the baseline using group values, substantially reducing training costs compared to PPO. Its impressive performance has revitalized interests in traditional RL frameworks.

**Limitation 4:** *overlooking reasoning diversity and efficiency.* Optimal reasoning trajectories are used to supervise the model's alignment. This *verbal alignment* is based on word-level overlap, encouraging the generated reasoning paths to resemble the optimal ones. Such approach fails to account for linguistic diversity of valid reasoning paths, which can varying in expression but still lead to the same outcome. Such an approach fails to capture underlying reasoning patterns, limiting the model's generalizability across scenarios. Moreover, using LLMs as judges to evaluate verbal reasoning trajectories incurs high computational costs from the frequent inferences (Chen et al., 2025b). Actionable insights are in Section 5: Action 4 and 5.

### 4.3. Evaluation and Regulation

A complete reasoning chain **A** is obtained after *Monitoring*. The *Evaluation and Regulation* module in Figure 5 leverages the feedback **O** to refinement. It is worth noting that while *Monitoring* acts as an ongoing observer of the thought process: "*thinking while doing*", i.e., the post-training phase aimed to enhance inherent reasoning capabilities. *Evaluation* and *Regulation* review the reasoning process as a whole: "*thinking after doing*", i.e., strategies applied during inference. Therefore, we focus on how existing feedback can help reasoning error correction methods during inference.

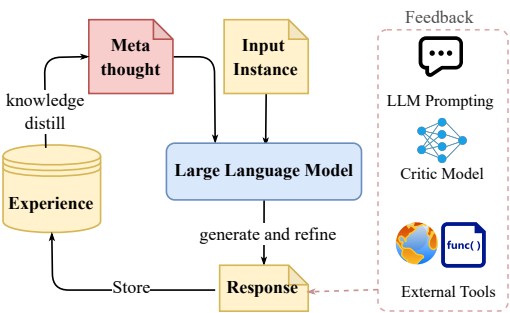

*Figure 5.* **Evaluation and Regulation**. Meta-thoughts from previous interactions are leveraged to enhance feedback. Other feedback sources are summarized in the dashed box.

### 4.3.1. OVERALL REASONING PROCESS EVALUATION

Existing typical feedback used to evaluate entire reasoning chains is summarized as: (i) prompting LLMs, (ii) leveraging trained critic models and (iii) integrating tools (shown in Figure 5). **LLM prompting**, however, does not guarantee that the underlying reasoning chain is faithful, accurate, or logically sound. **Task-specific critic models** for error detection, applied in areas like mathematical reasoning (Chen et al., 2024b), coding (Kumar et al., 2024), and logical reasoning (Tong et al., 2024), offer alternative approaches for improvement. To create training data for critic models, reasoning processes are first generated from a base LLM, then human annotators are asked to provide detailed annotations of error positions (which reasoning step) and error types, e.g., knowledge error or calculation error (Tyen et al., 2023; Uesato et al., 2022). It is worth noting that even if the critic model identifies errors successfully, it may still be unable to articulate them clearly. Other approaches **incorporate external tools**, such as *code compilers* (Shinn et al., 2023b; Yao et al., 2023b; Zhang et al., 2023a; Chen et al., 2024c), *Search engines* which allow LLMs to retrieve the latest evidence to ensure factuality (Varshney et al., 2023; Yu et al., 2023b; Peng et al., 2023), and *relation topology* analyzer for analyzing complex logical relations in graph-based systems (Zhang et al., 2023b).

The feedback generated by the aforementioned methods is predominantly sample-specific and fails to capture generalized error patterns across similar questions. More recently, thought templates (Yang et al., 2024; Wang et al., 2024g; Yang et al., 2025a), such as symbolized questions, have emerged as a viable approach to organizing similar questions. These templates facilitate evaluation across multiple analogous cases, thereby enabling feedback to reflect common reasoning patterns.

### 4.3.2. REASONING PROCESS REGULATION

Regulation concerns about error correction. Given the feedback produced in the evaluation phase, LLMs are expected to follow it and apply appropriate revisions through the so-

called self-reflection phase (Shinn et al., 2023a). However, there is no guarantee that LLMs will strictly follow the instructions. In fact, Yan et al. (2024b) observed that LLMs are often too stubborn to change their initial responses, even when provided with explicit feedback *"Your response is incorrect, please reconsider it."*. To explicitly guide LLMs to update their responses, TextGrad (Yuksekgonul et al., 2025) *backpropagates* the textual feedback provided by LLMs to refine the initial input prompts, effectively using natural language as a gradient. Another line of research trains LLMs using explicit error correction trajectories (Zhang et al., 2025a; Zelikman et al., 2024; Zeng et al., 2024a). DeepSeek-R1 also incorporates a similar 'think mode', typically involving a deliberate *wait*, to give the model more time to reflect on its previous thoughts, though it may also introduce overthinking issues (Chen et al., 2024d).

**Limitation 5:** *lack of adaptive meta-level error analysis.* Existing studies on refining LLMs primarily focus on instance-wise error detection and correction, which address errors for individual instances without leveraging insights from recurring error patterns (Yuksekgonul et al., 2025; Wang et al., 2024d). However, achieving broader improvements in LLM performance requires a meta-level approach that analyzes error patterns to identify underlying systemic issues or biases, thereby facilitating the development of strategies to prevent similar errors in future instances. While methods like the meta-buffer (Yang et al., 2024) and semantic-symbol prompts (Wang et al., 2024g) have shown the effectiveness of leveraging previous interactions and structured reasoning, they rely heavily on the inherent capabilities of LLMs and rigid, manually created prompt templates. Additionally, the thought templates may be suboptimal due to LLMs' limited instruction-following ability. Actionable insights are in Section 5: Action 3&4.

### 4.4. Meta-Reflection

Meta-reflection is the Bayesian learning process, designed to update the model's original views based on feedback across multiple tasks. It adopts a bi-level approach: first, the initial strategy $\mathbf{F}$ is refined, and then the meta-parameters $\Theta_{\mathbf{I}}$ and $\Theta_{\mathbf{E}}$ are optimized accordingly (the dash lines in Figure 2). The central challenge lies in efficiently consolidating the relationships among diverse tasks, ensuring that the meta-update remains optimal for all possible inputs.

MAML (Model-Agnostic Meta-Learning) (Finn et al., 2017) is a general framework for meta-learning, which learns a task-agnostic model initialization for quick adaptation to new tasks via bi-level optimization. To make this approach suitable for LLM deployment, several existing techniques can be adapted: (i) bi-level prompt optimization. (ii) modular training approaches such as LoRA. (iii) Bayesian inverse planning. Earlier methods such as MetaICL (Min et al., 2022) and MetaICT (Chen et al., 2022) avoid bi-level opti-

mization by training models on batches of diverse tasks in a continuous manner, simplifying the process to resemble conventional fine-tuning. Qin et al. (2023); Sinha et al. (2024) propose the **meta-prompt** and follow a bi-level optimization process. **Dynamic modular composition** (Huang et al., 2024a; Yang et al., 2025b), particularly when combined with LoRA, provides a flexible mechanism for recombining and reorganizing capability-specific components, enabling efficient generalization to new tasks through modular recombination. **Multi-agent RL** framework, such as ReMA (Wan et al., 2025) introduces a meta-thinking agent to coordinate among task-level agents, though they train the meta-level and task-level components separately rather than in an integrated manner.

To derive the optimized reasoning strategy $\mathbf{F}$ based on the cumulative reward $\mathcal{R}$, we can get inspiration from **Inverse planning**, which is to infer the agent's unobserved states, such as goals and beliefs, rooted in theory-of-mind (Shum et al., 2019). Specifically, we need to approximate $\mathcal{R}(\mathbf{F}; \mathbf{O}, \Theta_I, \Theta_E)$. One research direction leverages LLM preferences (Zhang et al., 2025c), for example, by using the model's generation logits given current observations and a candidate strategy for scoring. Alternatively, a Bradley–Terry model trained on preference data can serve as a proxy for the cumulative reward (Liu et al., 2024c).

**Limitation 6:** *lack of explainability and efficiency of meta-optimization.* While methods like LoraHub can decompose and recombine specialized capabilities for new tasks, they often suffer from safety and reliability issues (Hammoud et al., 2024; Hsu et al., 2024) during model merge. These risks highlight the limited understanding of the underlying mechanisms that govern model transfer learning and model parameter composition. Furthermore, there is a pressing need for efficient meta-optimization frameworks, such as multi-agent or multi-stage RL. Such approaches could serve as a foundational infrastructure for complex reasoning tasks, enabling better agent coordination, tool integration, and evolutionary skills for unseen tasks. Actionable insights are in Section 5: Action 5 & 6.

## 5. Actionable Insights

This section, driven by the limitations outlined in Section 4, presents potential future research directions.

**Action 1: Benchmark and metrics for meta-reasoning evaluation** To evaluate LLMs' meta-reasoning capabilities, well-defined benchmarks assessing self-awareness, introspection, and reflective reasoning are needed. Recent datasets like the SAD (Laine et al., 2024), AwareBench (Li et al., 2024b), and MM-SAP (Wang et al., 2024f) focus on introspection and multimodal reasoning, while MR-BEN (Zeng et al., 2024a) and MR-GSM8k (Zeng et al.,

2025) extend the evaluation to error analysis and qualitative insights. However, most of the existing benchmarks are on math and coding tasks, not generalized to wider reasoning tasks yet. Future work should aim to integrate these benchmarks into a unified framework and develop metrics beyond accuracy, such as calibration error, logical consistency, consistency rates, and generalization performance to assess meta-reasoning. For example, a very recent benchmark Feedbacker (Wang et al., 2025), provides a comprehensive evaluation framework to analyze the model's strengths and weaknesses via multifaceted feedback over various reasoning tasks, such as legal, ethical and causality.

**Action 2: Multi-view solvability with neuro-symbolic systems** While uncertainty or confidence scores can indicate capability-aware solvability, they are insufficient on their own. As discussed earlier, mission-aware solvability, such as rejecting unethical requests, must also be considered (Li et al., 2024b). The challenge lies in integrating these diverse solvability aspects into a unified decision-making framework. A neuro-symbolic approach could be explored, which combines symbolic reasoning's precision with neural modules' expressiveness (Andreas et al., 2016; Gupta et al., 2020). Different solvability metrics can be incorporated as neural modules, with the modular framework adapting to new metrics. The choice of a symbolic method to coordinate and execute neural modules is crucial: a probabilistic framework offers robustness (Nafar et al., 2024), while a logic-based method ensures precision (Servantez et al., 2024), depending on the task's priorities.

**Action 3: Adaptive reasoning strategy generation** Current "plan-to-plans" methods typically generate a single strategy for a given reasoning instance or for all instances within a task (Zou et al., 2023; Gao et al., 2024b). However, tasks may require multiple reasoning skills, such as knowledge retrieval and numeric calculation. To enable LLMs generalize across various tasks, we could map input context representations into a latent concept space, where each concept corresponds to a distinct reasoning skill. Solving a task would involve identifying relevant skills and generating answers conditional on them. The Mixture-of-Expert (MoE) framework allows dynamic allocation of reasoning skills (experts) to specific instances. Recent work combining MoE with parameter-efficient fine-tuning has enhanced the efficiency. Furthermore, hierarchical MoE could further improve skill sharing across tasks (Li et al., 2025b). Another possible way is to utilize Bayesian inverse planning (Wu et al., 2021; Zhang et al., 2025c) by treating the reasoning skills as the latent variable between meta knowledge and reasoning actions. By observing the results of reasoning actions, the posterior of the reasoning skills can be updated by the Bayes rule (Tonolini et al., 2024), enabling an adaptive reasoning skill selection.

**Action 4: Self-play for meta-rewards seeking** Human intelligence develops multifaceted self-assessment for reasoning and dynamically introduces new criteria through interactions with the environment. In contrast, current reward systems for reasoning monitoring are monofaceted, focusing primarily on correctness and remaining static, making it difficult to adapt to evolving distributions in policy models (Tao et al., 2024). Therefore, we propose leveraging multifaceted and dynamic meta-rewards through self-play. This claim aligns with recent theoretical and empirical findings that scaling feedback/rewards could lead to significant improvements in both training and inference phases (Snell et al., 2024; Wu et al., 2024c). Such a self-evolutionary system enables LLMs to autonomously acquire, refine, and learn from self-generated experiences or nuanced internal signals, such as confidence (Zhao et al., 2025). Additionally, a self-play paradigm reduces the reliance on human preference data and mitigates reward hacking issues. Achieving this paradigm requires algorithms that provably converge to the Nash equilibrium of two-player constant-sum games (Wu et al., 2025) and fully exploit the internal feedback during the interaction process.

**Action 5: Latent-space reasoning for better diversity and efficiency** Most existing reasoning approaches generate explicit verbal intermediate steps in an autoregressive manner. However, errors in these steps can accumulate, leading to cascading mistakes, challenges in self-correction, and inefficiencies (Deng et al., 2024; Lin et al., 2021; Ye et al., 2024). By internalizing explicit thoughts into a latent space, we can capture reasoning patterns independent of linguistic style, encourage the model to think more and talk less, as well as avoid unnecessary cost on generated lengthy sequences to accelerate the model inference speed at the same time. Preliminary work (Deng et al., 2024; Hao et al., 2024; Shen et al., 2025) has shown promise in faster inference by completely bypassing lengthy intermediate verbal steps, though still lagging behind the verbalized CoT approaches. Instead of leveraging additional tokens, looped transformers are also promising as they enhance the thinking via leveraging additional depth for more computation to reason, which can also be interpreted as a form of latent reasoning (Geiping et al., 2025; Saunshi et al., 2025; Yu et al., 2025). Additionally, A well-regularized latent space can further improve interpretability and global controllability (Ye et al., 2024; Su et al., 2025), as well as accelerate the simulation process via embedding search (Chen et al., 2025b). An appropriate manipulation in the latent space can also encourage the exploration for better math reasoning (Zhu et al., 2025; Zhang et al., 2025b) and retrieved-augmented QA (Hu et al., 2025).

**Action 6: Interpretable and efficient training for meta-knowledge consolidation** To enhance the adaptability and efficiency of large models, it is crucial to identify and leverage the distinct roles of different subnetworks or skill-specific agents/tools, allowing selective leveraging/updating of the most relevant components for particular inputs. This targeted approach improves the model's ability to understand LLMs' knowledge learning and consolidation process. Recent work shows that large performance gains often come from updating 5%-30% of model parameters (Mukherjee et al., 2025). Therefore, mechanistic interpretability can provide valuable insights into rigorous causal effects (Yan et al., 2024a; Bereska & Gavves, 2024) linking specific model components to outputs. Moreover, equipping agents with the capability to recognize and understand their own knowledge boundaries (Qiao et al., 2025) is crucial in multi-objective cooperative frameworks (), in which a meta-level coordinator oversees the collaboration among multiple agents. Such self-awareness enables agents to contribute more effectively, fostering robust collaboration and reducing the risk of conflicts or redundant efforts.

# 6. Alternative Views

Some suggested that LLMs should operate under human oversight to ensure robust reasoning and decision-making (Rafailov et al., 2023). Others argued that robust reasoning can be achieved by integrating structured knowledge bases and symbolic reasoning into LLMs (Hao et al., 2023b; Shu et al., 2024). Our proposed LLM meta-reasoning framework may also face criticism for adding complexity and computational overhead. Our arguments are: (i) Human oversight is resource-intensive and impractical to scale for every use case, especially in real-time applications. (ii) Symbolic reasoners alone struggle with the complexity and nuance of natural language. External solvers, including symbolic reasoners, are part of our meta-reasoning framework. (iii) Unlike task-specific models, meta-reasoning allows LLMs to generalize better across unfamiliar tasks by reflecting on and adapting their reasoning strategies. Instead of fine-tuning separate models for every domain, we can adapt dynamically, reducing development time and computational costs in the long run.

# 7. Conclusion

We introduced a Bayesian Meta-Reasoning framework that integrates key components such as self-awareness, monitoring, evaluation, and meta-reflection, inspired by human cognitive processes. It addresses fundamental limitations in existing approaches, such as a lack of dynamic adaptability, limited variety in reasoning pathways, and inefficiencies in task-specific updates. By incorporating external resources, meta-knowledge-based assessments, and flexible sampling mechanisms, our approach could show significant promise in complex, unstructured reasoning across domains. Furthermore, we highlighted key challenges in meta-reasoning and proposed potential future research directions.

## Acknowledgments

This work was supported by the UK Engineering and Physical Sciences Research Council (EPSRC) through a Turing AI Fellowship (grant no. EP/V020579/1, EP/V020579/2).

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

# A. Appendix: Litereature Review

This appendix presents a brief literature review of existing studies that offer insights relevant to components of our meta-reasoning framework, including meta-reasoning in cognitive science and machine intelligence, uncertainty estimation and calibration, LLM reasoning with reward models, and LLM refinement through feedback.

## A.1. Meta-Reasoning

meta-reasoning, the process of thinking about one's own thinking, is of significance in both human and artificial intelligence (Cox & Raja, 2011; Ackerman & Thompson, 2017).

**Meta-Reasoning in Cognitive Science** Meta-reasoning in cognitive science involves several key theories explaining how individuals monitor and regulate their reasoning processes. Dual-Process Theory (Kahneman, 2011) suggests that reasoning involves intuitive and deliberative systems, with meta-reasoning influencing how individuals balance these systems. The Metacognitive Reasoning Theory (Koriat, 2000) highlights the "Feeling of Knowing", which guides when further cognitive effort is required. The Diminishing Criterion Model (Ackerman, 2014) suggests that confidence declines over time, leading individuals to accept answers with lower confidence. Judgment of Solvability (Lauterman & Ackerman, 2019) focuses on how individuals assess whether a problem is solvable, influencing their persistence or decision to abandon the task. "Feeling of Errors" (Gangemi et al., 2015) explores how people detect errors and adjust their reasoning accordingly. These theories highlight how metacognitive processes control cognitive effort and shape decision-making. Grounded in the seminal framework proposed by Nelson (1990) for monitoring learning and memory, Ackerman & Thompson (2017) proposed a meta-reasoning framework, which consists of two main components, *metacognitive monitoring* and *metacognitive control*. *Metacognitive monitoring* involves subjective assessment of the likelihood of success or failure in a particular cognitive task, and guide decisions on action, time, and effort allocation. *Metacognitive control* determines the initiation, termination, or adjustment of the allocation of metal effort to a cognitive task.

**Meta-Reasoning in Decision-Making and Multi-Agent Systems** Cox & Raja (2011) discussed meta-reasoning from both AI and cognitive science perspectives, structured around a central model where meta-reasoning controls and monitors reasoning, guiding decisions on when to act or continue thinking. In AI research, meta-reasoning has been extensively studied in search and planning. For example, Lin et al. (2015) focused on the general meta-reasoning decision problem, which involves balancing the cost of planning with the quality of the resulting actions to maximize an agent's long-term utility. They proposed approximate algorithms within Markov decision processes (MDPs) using the Bounded Real-Time Dynamic Programming technique to evaluate the computational value of further reasoning without relying on prior domain-specific data. Van Zee & Icard (2015) focused on one aspect of meta-reasoning, which is reconsidering or adjusting plans as new information becomes available. They explored optimal strategies for when an agent should 'think' (re-plan) versus 'act' under changing conditions, suggesting that flexible and meta-level strategies can enhance decision-making across diverse environments. Elboher et al. (2023) studied situated temporal planning and proposed algorithms for concurrent action execution and deliberation. There are different aspects of meta-reasoning, such as using meta-level control in decision-making and multi-agent systems (Raja & Lesser, 2007; Cheng et al., 2013; Langlois et al., 2020), introspective monitoring for self-improvement (Murdock & Goel, 2008; Cox, 2011; Toy et al., 2024), models of self-awareness in cognitive agents (Samsonovich et al., 2008; Proust, 2013; Samsonovich & De Jong, 2013; Chatila et al., 2018; Subagdja et al., 2021), and using metacognitive reinforcement learning to understand how to think (Krueger et al., 2017).

**Meta Reasoning for Language Models** Most recent work conducts prompt-engineering on the frozen LLMs for so-called meta-reasoning. Wang et al. (2024h) adapted meta-prompt to identify the specific task before conducting the sample-level reasoning. Yang et al. (2024) distilled the knowledge from multiple instances and their solutions into thought template saved in a buffet, which will be accessed before instantiated Reasoning. Wang et al. (2024g) deconstructed reasoning-independent semantic information into generic symbolic form and therefore transform various questions into meta-from. Performances improvements are observed in arithmetic, symbolic, and logical reasoning and multi-agent mental gaming. Some studies claimed that task-agonistic instructions help LLMs learn to think. Wu et al. (2024b) prompted LLMs to generate thoughts before its response via feeding a thought template filled with human instruction, then used the self-generated responses for direct preference optimization (DPO). The deliberate thought, thought hidden from the end user, is supposed to be part of the internal thought of LLMs, making LLMs perform better in general reasoning tasks.

**The gap between machine meta-reasoning and human intelligence** Despite the advancements discussed, existing meta-reasoning approaches in LLMs remain limited in capturing the depth and complexity of human cognition. While

methods such as meta-training, prompt-engineering, and instruction-tuning have demonstrated efficacy in task-specific generalization and structured reasoning, they often rely on rigid templates or narrowly defined optimization frameworks. These strategies lack the ability to dynamically adapt to diverse, unstructured contexts or to introspectively evaluate their reasoning processes—core aspects of human meta-reasoning. For instance, the ability to reflect on one's knowledge state ("*knowing what one knows*") and detect gaps or uncertainties (Flavell, 1979). This extends to the awareness of errors (Gangemi et al., 2015), where humans can identify and correct mistakes through introspection and feedback processes.

### A.2. Uncertainty Estimation and Calibration

In the meta-reasoning process of humans, one will first have an initial judgment of solvability on the given task to prepare for the assessment of knowledge and strategies. Similarly, in the meta-reasoning process of LLMs, an LLM should also assess its initial judgment of the solvability and difficulty of the given task, which can be achieved by estimating the confidence score or uncertainty score of the LLM's generation. To be more specific, there are two types of uncertainty estimation methods. Uncertainty calibration further refines these estimates by improving their reliability.

**Real-Time Uncertainty Estimation**   The first type of method is real-time uncertainty estimation, where the LLMs generate the output and its uncertainty simultaneously (Geng et al., 2024). *Linguistic-based methods* prompt LLMs to express uncertainty in human language, which assumes LLMs are well-calibrated with verbalized confidences, i.e., LLMs can express their uncertainty towards output either in numeric expression (e.g., 0-1) or in linguistic expression (e.g., certain, likely, no chance) (Tian et al., 2023; Mielke et al., 2022). *Logit-based methods* estimate the sentence uncertainty by token-level entropy (Huang et al., 2025). To incorporate semantics, Duan et al. (2023) introduced the concept of token-level relevance, which evaluates the relevance of the token by comparing semantic change before and after moving the token with a semantic similarity metric. Then, sentence uncertainty can be adjusted based on the token's relevance.

**Post-hoc Uncertainty Estimation**   The second type of method is the post-hoc uncertainty estimation, where the uncertainty is estimated after the output has been generated. *Consistency-based methods* assume that when the LLM is certain about a given concept, the sampled responses are likely to be similar and contain consistent facts, while for hallucinated facts, stochastically sampled responses are likely to diverge and may contradict one another. Manakul et al. (2023) proposed to sample multiple generations based on one input and calculate the similarity score between the target and generations, then the similarity scores are aggregated and taken as the uncertainty towards the target. *Distribution-based methods* convert the LLM outputs into embeddings and estimate the output uncertainty based on the distribution of the embeddings. Catak & Kuzlu (2024) proposed a novel geometric approach to uncertainty quantification using convex hull analysis, which leveraged the spatial properties of response embeddings to measure the dispersion and variability of model outputs. Kuhn et al. (2023) proposed semantic entropy, an entropy that incorporates linguistic invariance created by shared meanings.

**Uncertainty Calibration**   Uncertainty calibration aims to align confidence scores with actual correctness to improve prediction reliability. *Supervised-based methods* fine-tune LLMs on datasets containing both correct and incorrect answers along with their uncertainties to improve the model's ability to estimate uncertainty (Liu et al., 2024b; Kapoor et al., 2024). *Prompting-based methods* leverage prompt augmentation techniques, such as paraphrasing or option permutation, to create ensembles that enhance calibration without additional training (Jiang et al., 2023; Xiong et al., 2024b).

### A.3. LLM Post-training with Reward

There are currently two training paradigms to enhance LLMs to reason after pre-training: Supervised Finetuning (SFT), or Imitation Learning, and Reinforcement Learning (RL). SFT allows the model the finetune on annotated reasoning chains to learn the reasoning pattern. While RL, the current SOTA method, requires a reward for the LLM to maximize, and the LLM learns the reasoning patterns via its own explorations.

**Reward Modelling**   In the LLM framework, reward models are generally categorized into two types: Outcome Reward Models (ORMs) and Process Reward Models (PRMs). ORMs primarily evaluate completed outputs, and rule-based rewards applied to entire outputs—though not learned—can also be regarded as a form of ORM. PRMs, by contrast, have shown effectiveness in both inference (Lightman et al., 2024) and training (Wang et al., 2024c). Recently, with the release of Deepseek-R1, rule-based ORMs have gained increasing attention as a practical method for post-training policy models.

A significant limitation of PRMs is their reliance on high-cost human annotations for CoT paths. To address this, methods such as Monte Carlo (MC) estimates have been employed to quantify the quality or value of each reasoning step (Wang et al.,

| Methods | Process or Outcome | Reward Type | Ex(Im)Plicit | Training Algorithm |
|---|---|---|---|---|
| Llama3-Instruct Models (Grattafiori et al., 2024) | ORM | Preference | Explicit | SFT+DPO |
| Qwen2.5-Instruct Models (Qwen et al., 2025) | ORM | Preference | Explicit | SFT+DPO+GRPO |
| AlphaMath (Chen et al., 2024a) | PRM | Correctness | Explicit | SFT |
| Math-Shepherd (Wang et al., 2024c) | PRM | Correctness | Explicit | SFT+PPO |
| Chain of Preference Optimization (Zhang et al., 2024c) | PRM | Correctness | Explicit | SFT+DPO |
| DeepSeek-R1-Zero (DeepSeek-AI et al., 2025) | ORM (Rule-based) | Correctness, Format | Explicit | GPRO |
| DeepSeek-R1 (DeepSeek-AI et al., 2025) | ORM (Rule-based) | Correctness, Format | Explicit | SFT+GPRO |
| EBRM (Lochab & Zhang, 2025) | ORM | Energy | Implict | SFT+PPO |

*Table 2.* Feebback signals used for LLMs post-training (RL-based).

2024c; Chen et al., 2024a). Additionally, due to the strong in-context learning capabilities of LLMs, the "LLM-as-a-judge" approach (Yao et al., 2023a; Zhang et al., 2024a) has become a popular alternative to replace PRMs. Recently, (Zhang et al., 2025d) advanced this approach by integrating MC estimation with LLM-as-a-judge, establishing a new state-of-the-art (SOTA) on the PRM benchmark (Zheng et al., 2025).

**Search with Rewards**   Once suitable rewards are identified, they can be leveraged to control the LLM's reasoning process generation. A straightforward and intuitive method for incorporating rewards is Best-of-N (Lightman et al., 2024), which selects the highest-scored generation from a set of candidates. PRMs further enable fundamental tree search algorithms, such as Depth First Search (DFS) and Breadth First Search (BFS) (Yao et al., 2023a). Although DFS is rarely used due to its limited exploration capabilities, BFS is frequently extended into Beam Search (Yao et al., 2023a; Xie et al., 2023), offering a balance between search quality and computational efficiency.

For more computationally demanding scenarios, Monte Carlo Tree Search (MCTS) can be used in conjunction with PRMs (Hao et al., 2023a; Wan et al., 2024). Interestingly, as shown by (Snell et al., 2025), the more complex MCTS often underperforms the simpler Beam Search, which in turn only outperforms Best-of-N under low computational budgets. This counterintuitive trend is often attributed to reward over-optimization, where methods are misled by imperfect reward signals (Qin et al., 2024). Despite these challenges, MCTS has shown promise on more challenging questions. Moreover, (Zhang et al., 2024a) demonstrated that integrating MCTS with a self-refine module enabled an 8B model to achieve performance comparable to GPT-4 Turbo, highlighting the potential of advanced search algorithms like MCTS.

**Training with Rewards**   Rewards can also be leveraged to further improve the performance of policy models. Popular instruction-finetuned models such as LLaMA3 (Grattafiori et al., 2024) and Qwen2.5 (Qwen et al., 2025) follow a two-stage training process: they are first finetuned on annotated QA datasets, and then refined using preference data via DPO. The main distinction lies in the online reinforcement learning stage—Qwen2.5 employs GRPO, while LLaMA3 continues with DPO.

Models like AlphaMath, Mathshepherd, and Chain of Preference Optimization utilize PRMs to assign values to intermediate reasoning steps, though their training strategies vary. Empirical evidence suggests that reinforcement learning typically outperforms simple rejection sampling.

A recent breakthrough, Deepseek-R1-Zero (DeepSeek-AI et al., 2025), achieves state-of-the-art (SOTA) performance by entirely skipping the supervised finetuning (SFT) stage. Instead, it trains a model from scratch using a pure RL method (GRPO) with only two simple reward signals: a correctness reward, which checks whether the final answer is correct, and a format reward, which ensures the answer is presented in the correct format. Despite the simplicity of these signals, the model achieves performance comparable to OpenAI-o1 (OpenAI, 2024), the leading reasoning model at the time.

However, Deepseek-R1-Zero tends to produce reasoning chains with poor readability, as the rewards do not explicitly encourage clarity. To address this, the authors also release Deepseek-R1 (DeepSeek-AI et al., 2025), which includes a finetuning stage before RL and achieves similar performance with significantly better readability.

Lastly, EBRM explores the use of implicit rewards for training, demonstrating improved robustness and generalization capabilities. This approach opens the door to leveraging rewards that go beyond human-interpretable signals.

### A.4. LLM Refinement During Inference

Refining the reasoning process of LLMs using feedback has emerged as a promising approach to enhance their performance, reliability, and adaptability across diverse tasks. Feedback enables iterative improvements in reasoning by providing explicit information about errors and areas requiring adjustment.

**Types of Feedback**  Existing feedback are falls into two categories, scalar and verbal. *Scalar feedback* can be Boolean values (e.g., integers 0 or 1), or consistency scores (e.g., a decimal probability) (Wang et al., 2023a; Fu et al., 2023a; Pan et al., 2024), which is observed to be highly correlated with correctness (Yao et al., 2023a; Li et al., 2023; Yan et al., 2024b). *Verbal feedback* is more informative and explainable. The most direct way of generating feedback is to *prompt the LLMs to* assess their current reasoning process (Lu et al., 2023; Liang et al., 2024). However, recent studies (Huang et al., 2024b; Li et al., 2024c; Yan et al., 2024b) observed that LLMs often fail to provide reliable feedback due to their limitations in knowledge grounding. Researchers then *retrieve external knowledge or incorporate external tools*, such as code compiler to improve the validity of verbal feedback (Shinn et al., 2023b; Yao et al., 2023b). Verbal feedback from multi-agents or multi-perspective is also shown to be more robust (Fu et al., 2023b; Bo et al., 2024; Zhang et al., 2024b).

| Feedback Category | Representative Papers |
|---|---|
| Template-based feedback | Self-Refine (Madaan et al., 2023), PromptAgent (Wang et al., 2024d), LLMsCF (Tyen et al., 2023), LargeLM (Huang et al., 2024b) |
| Critic model feedback | Step-Level Preference (Chen et al., 2024b), Self-Correct (Kumar et al., 2024), Learning from Mistakes (Tong et al., 2024), Math Error Localization (Uesato et al., 2022), REFINER (Paul et al., 2024), DARS (Li et al., 2025a) |
| Token-based backtracking | Backtracking (Zhang et al., 2025a), QuietStar (Zelikman et al., 2024) |
| Tool-assisted feedback | ***Code Interpreter***: Self-Edit (Zhang et al., 2023a), Self-Debug (Chen et al., 2024c), CRITIC (Gou et al., 2024), ReTool (Feng et al., 2025a), ToolRL (Qian et al., 2025); ***Search Engines***: ReAct (Yao et al., 2023b), CheckFacts (Peng et al., 2023), ASI (Varshney et al., 2023), Search R1 (Jin et al., 2025), R1 Searcher (Song et al., 2025); ***Logic*** and *relation topology*: Logic/Graph Analyser (Zhang et al., 2023b) |

*Table 3.* Categorization of feedback techniques for refining LLM reasoning.

**Refining Reasoning Process with Feedback**  To update the original reasoning process, we can directly prompt the LLMs with feedback (Madaan et al., 2023; Yan et al., 2024b; Bo et al., 2024), train a separate critic model, introduce explicit backtrack tokens and utilize external solvers.

(i) As much of the feedback is template-based (Huang et al., 2024b; Tyen et al., 2023), they typically offers limited information about the error. Wang et al. (2024d) then proposed *PromptAgent* to dynamically optimize the initial prompt template based on feedback. Specifically, a base LLM is used to collect errors from samples and the optimized LLM (usually superior to the base LLM) is then used to offer error feedback and update the initial prompt. Additionally, multi-agent framework enable the multi-faceted feedback for complex reasoning tasks, such as scientific paper replication (Xiang et al., 2025b).

(ii) Many task-specific critic models are trained for error correction in math (Chen et al., 2024b), coding (Kumar et al., 2024), QA and logic reasoning (Tong et al., 2024) tasks. To create the training data for critic models, they first sample LLM-generated reasoning processes from a base model and ask human annotators to provide detailed annotations of error positions (which reasoning step) and error types, e.g., knowledge error or calculation error (Tyen et al., 2023; Uesato et al., 2022).

(iii) Some studies abandon the separate critic models and instead introduce special tokens, such as *[RESET]* to trigger explicit error correction processes (Zhang et al., 2025a; Zelikman et al., 2024).

(iv) Tools like a *Code Interpreter* enable LLMs to validate reasoning and refine outputs based on compiler results (Feng et al., 2025a; Qian et al., 2025), applied in mathematical and logical reasoning tasks (Yao et al., 2023b; Zhang et al., 2023a; Chen et al., 2024c). *Search engines* allow LLMs to retrieve the latest evidence to ensure factual correctness and validity (Varshney et al., 2023; Yu et al., 2023b; Peng et al., 2023; Jin et al., 2025; Song et al., 2025). *Logic* and *relation topology* analyzer assists in analyzing complex logical relations and dependencies in graph-based reasoning systems (Zhang et al., 2023b).

