# OpenReview forum: "Position: LLMs Need a Bayesian Meta-Reasoning Framework for More Robust and Generalizable Reasoning"
_ICML.cc/2025/Position_Paper_Track — ICML 2025 Position Paper Track poster_

### Official Review · Reviewer_1XTN · 2025-03-04

**Significance:** 3
**Argument Clarity:** 2
**Rating:** 3
**Confidence:** 4

**Questions:**

1. Line 378: Why Meta prompts serve as task-specific templates and are customized for individual samples? This seems to contradict with section 4.1.1.
2. Line 436: What do external resources, meta-knowledge-based assessments, and flexible sampling mechanisms mean? Please explain these methods in details.
3. Typo: "flexiable" in line 436.

**Discussion Potential:**

3

**Paper Summary:**

The paper argues for a position that LLMs need a bayesian meta-reasoning framework for more robust and generalizable reasoning.
To demonstrate this perspective, the paper outlines several open problems concerning LLM reasoning and proposes a conceptualized framework with five components to illustrate how to address the meta-reasoning in LLMs.
By reviewing recent studies relevant to each of the components, the paper examines limitations in current work on improving meta-reasoning in LLMs.
Finally, the paper lists several perspectives to inspire future work to address the challenges of meta-reasoning in LLMs.

## update after rebuttal
The topic the paper discusses is timely and important, but the proposed meta-reasoning framework lacks of empirical or theoretical evidence to validate its feasibility.

**Position:**

Yes

**Position In Title:**

Yes

**Related Work:**

3

**Strengths And Weaknesses:**

Strengths:
1. Enhancing meta-reasoning is a critical next step for improving LLM reasoning capabilities, and it is likely to inspire discussion and future work on more robust and generalizable reasoning of LLMs.

2. The paper is well-written and informative, and its position is well supported by related work.

Weaknesses:
1. The limitations argued in section 4 are not accompanied by corresponding potential solutions (better in details and case studies) to address these issues. In addition, the connections between the limitations in section 4 and the future directions in section 5 are not clearly illustrated. Perhaps it can be better to provide actionable insights that correspond to the limitations.

2. The paper shows an example about how to use the proposed meta-reasoning framework for scientific hypothesis generation at the end of Section 3. However, it is only a tentative plan, and perhaps it is better to instantiate this plan to provide solid evidence about whether the meta-reasoning framework can improve scientific reasoning in LLMs.

Suggestions:
1. Please state the position in bold in the introduction part.

**Support:**

2

---

> ### Author Rebuttal · Authors · 2025-03-29
>
> Thanks for your constructive suggestions and questions!
>
> **Q1: Correspondence between Limitation and actionable insights**
>
> Thank you for this valuable suggestion. We will explicitly highlight the correspondence between identified limitations and our proposed actionable insights in Section 5 to improve clarity and readability.
>
> **Q2: Instantiate the Scientific reasoning application**
>
> An example of scientific hypotheses generation.
>
> Initialisation:
> - I (Internal View): LLM’s parametric knowledge, e.g., general biomedical knowledge.
> - E (External View): Task-specific knowledge, such as retrieved relevant information from domain-specific knowledge bases， e.g., genomic databases.
>
> Bayesian Learning:
> - Updating reasoning strategies at the task-level.
> - Self-awareness to initiate the distribution of reasoning strategies, such as cross-domain analogy (identify analogous phenomena across fields), constraint satisfaction (formulate hypotheses that meet known constraints).
> - Choose an initial reasoning strategy F (e.g., cross-domain analogy)  informed by foundational and task-specific knowledge.
> - Generate reasoning process A using the current F, such as exploring Alzheimer’s disease using analogies from cancer research to generate novel hypotheses.
> - Follow A to generate hypotheses.
> - Evaluate the quality of hypotheses based on novelty, feasibility, plausibility,  and record feedback (O).
> - Update F using $p(F|O, I, E) \propto p(O|F, I, E)p(F|I, E)$
> - Updating knowledge priors at the meta-level.
> - Update knowledge priors I and E based on meta-rewards computed across multiple hypotheses generation tasks (e.g., Over multiple hypothesis-generation cycles in different neurodegenerative diseases, the model recognises a pattern that Single Nucleotide Polymorphism affecting microglial activation tend to be highly relevant. The LLM updates its priors to prioritise genes involved in neuroinflammation across future tasks).
>
> Bayesian Inference:
> - Generate hypotheses based on the chosen reasoning strategy
> - Evaluate the quality of hypotheses based on novelty, feasibility, plausibility, recorded as feedback O
>
> **Q3: Meta Prompts?**
>
> In line 378, the meta prompts refer to the bi-level prompts proposed in [1,2] in the mix-task scenarios. So the high-level meta-prompt is adopted to specify the targeted task (such as “To solve this problem, we can use Arithmetic sequence sum formula”), here is the Arithmetic question, and the lower level prompt is targeted to provide more details of the individual arithmetic question.  Could you please specify where it might contradict in Section 4.1.1 in case this answer doesn't fully address your concern?
>
> **Q4:  External resources, meta-knowledge-based assessments, and flexible sampling mechanisms**
> - External resources refer to the integration of external tools to enhance reasoning capabilities, retrieving external knowledge from knowledge base, etc.
> - Meta-knowledge-based assessment corresponds to our monitoring process using meta-rewards, which adaptively guide the reasoning process without ground truth, and beyond the task-specific signal.
> - The flexible sampling mechanism aligns with reasoning in the latent space, allowing for more diverse and robust exploration during inference, such as GFlowNet[3].
>
> **Q5: Typos and other suggestions.**
>
> Thanks for pointing out! Will fix the typos and highlight the *complete sentence* of position in the introduction.
>
>
> Please let us know if you have further concrete questions or concerns! Thanks for your engagement with our work!
>
> **References**
>
> [1]Strategic chain-of-thought: Guiding accurate reasoning in llms through strategy elicitation
>
> [2]Meta-cot: Generalizable chain-of-thought prompting in mixed-task scenarios with large language models
>
> [3]Gflownet foundations

---

> > ### Comment · Reviewer_1XTN · 2025-04-02
> >
> > Thanks for your response. However, answers for Q1 and Q2 do not address my concerns. For Q1, please state in detail that how you will strengthen the connections between Section 4 and Section 5. This is critical to the quality of the paper. For Q2, authors have misunderstood my requests (possiblly due to my ambiguous wording). Actually, I want to ask for a proof-of-concept experiments or preliminary experiments to support the framework (as also suggested by Reviewer oVhh) to demonstrate its feasibility and usefulness in empirical studies. Often times, a complex framework or system in theory does not work in real-world applications, such as the Beyasian learning methods not scalable to complicated scenarios. Authors' response to Reviewer oVhh is also not convincing and many position papers in ICML do include experimental validations to support their position. In addition, explanations for Q3 and Q4 should be included in the revised paper. At least for now, I do not think this paper could be published on ICML.

---

> > > ### Author Response · Authors · 2025-04-05
> > >
> > > ### **Q1: Connections between section 4 and section 5**
> > >
> > > To enhance the connections between section4 and section5, we revise section 5 as follows:
> > >
> > > > **Benchmark and metrics for meta-reasoning evaluation.** *It is to provide an overall evaluation framework to evaluate the multi-view framework for task solvability (limitation1), the modal adaptively in incorporating meta-level thought (limitation 5), and conduct meta-level error analysis (limitation 6).*
> > >
> > > > **Multi-view solvability with neuro-symbolic systems.**  *Our self-awareness module integrates multifaceted solvability to enhance reasoning diversity, addressing the **lack of a multi-view framework for solvability** (Limitation 1).*
> > >
> > > > **Adaptive reasoning strategy generation.**  *By consolidating multiple latent skills, our approach enables adaptive strategy selection, tackling the **lack of adaptability in latent skill selection** (Limitation 2)*
> > >
> > > > **Self-play for meta-rewards seeking.** *It can overcome the **shortcomings of existing static and oversimplified reward function** (limitation 3).*
> > >
> > > > **Latent-space reasoning for meta-cognitive LLMs.** *By internalising explicit thought processes into a latent space, we improve both **sample diversity** and reasoning efficiency (Limitation 4)*
> > >
> > > >  **Modular training for meta-knowledge consolidation.** *We structure latent skills in a modular format, enabling the efficient construction of meta-level thoughts by recombining these latent skills for unseen tasks (Limitation 7: **adaptability for meta-level templates**), it also enhance the **explainability and efficiency of meta updates**. (Limitation 8)*.
> > >
> > > ---
> > >
> > > ### **Q2: Proof-of-concept experiment and feasibility of Bayesian learning**
> > > Sorry for the misunderstanding of your request, we have updated a preliminary experiment on prompting ChatGPT (suggested by reviewer oVhh) as part of the response to reviewer oVhh (due to word limit).
> > >
> > > For your particular concern of feasibility, such as the Bayesian learning, we use Bayesian inference to estimate the prior distribution of the reasoning strategy $\textbf{F}=(f_1, f_2,...,f_N)$.
> > > > A specific and simple reasoning strategy (only one element, $f1$) is the inference strategy, direct answer or Chain-of-Thought (CoT). We integrate Bayesian learning into uncertainty estimation (a pratical solution can be found in [1]), and If the estimated uncertainty is very low—indicating that the question is relatively simple—the model can directly generate an answer without requiring CoT. So, we can add a context along with the question, *please answer the question directly".
> > >
> > > In our context, Bayesian inference serves to generate a meta-level reasoning strategy $\textbf{F}$, which acts as an additional meta-level input to enhance the LLM’s self-awareness. Consequently, the token generation process is no longer based on $p(x_{t}|x_{<t})$, it should be $p(x_t|x_{<t};\textbf{F})$.
> > >
> > > With such additional inputs, [2] has already introduced a practical framework for such a constrained optimisation problems of the form $p(O|f), \forall f \in F, f= (f_1, f_2,...,f_N)$ shown in Section 3.
> > >
> > > ---
> > >
> > > ### **Q3 and Q4: add explanations into revised paper**
> > >
> > > Sure, these clarification can enhance our clarity of our paper. So, we will add the detailed description of the meta-prompt in section 4 in our revised paper; Also, we will add the additional explanations of *External resources*, *meta-knowledge-based assessments*, and *flexible sampling mechanisms* into the conclusion part.
> > >
> > > ---
> > >
> > > Please let us know if you have further concrete questions or concerns that we can address. Thank you.
> > >
> > >
> > >
> > > **Reference:**
> > >
> > > [1] Bayesian Prompt Ensembles: Model Uncertainty Estimation for Black-Box Large Language Models
> > >
> > > [2] Bird: A Trustworthy Bayesian Inference Framework For Large Language Models

---

### Official Review · Reviewer_CRfQ · 2025-03-10

**Significance:** 3
**Argument Clarity:** 4
**Rating:** 4
**Confidence:** 4

**Questions:**

See Strengths And Weaknesses

**Discussion Potential:**

3

**Paper Summary:**

This paper concerns a very important topic -- improving foundation models' generalizability and robustness based on inspirations from cognitive science. It identifies key challenges for foundation models, i.e. the  robustness of inference and the generalization of learning. It points out issues with the current  foundation model frameworks:
1) for  SFT+RL the need to adapt to new reasoning tasks.
2) for  iterative refinement and prompt optimization the need for efficiency and generalizability.

A few design goals are identified based on cognitive science theories
1) self-awareness (or explicit memory)
2) selection of different strategies
3) long-term planning with generalizable patterns
4) efficiently learn new knowledge

A Bayesian inference and learning framework is sketched.

**Position:**

Yes

**Position In Title:**

Yes

**Related Work:**

4

**Strengths And Weaknesses:**

I think this is a very timely paper. Even though the specific framework might not be realistic, the identified issues and design goals provide valuable guidance to future research.

**Support:**

4

---

> ### Author Rebuttal · Authors · 2025-03-29
>
> Thanks for your acknowledgement!

---

### Official Review · Reviewer_8n3F · 2025-03-11

**Significance:** 2
**Argument Clarity:** 2
**Rating:** 2
**Confidence:** 5

**Questions:**

1. Metareasoning has been a part of computer science/artificial intelligence research for the last several decades. Let me provide just one, probably biased, citation: https://dl.acm.org/doi/10.5555/3020652.3020691, which however cites many relevant works and is cited by followups. Does the framework of rational metareasoning suit the needs of your proposed approach?

2. In connection with the above, here is a work https://arxiv.org/abs/2410.05563 (the same as in the strengths/weaknesses, the link is to the paper) that seemingly develops rational metareasoning framework for LLMs. What is missing in this work? How is this work related to your position?

3. You propose a Bayesian framework in terms of straightforward Bayesian modeling, and connect it with reinforcement learning. A modern tool for tackling similar problems is causal reasoning, which you do not seem to mention. What makes you think that pure Bayesian inference rather than causality calculus, is the right choice?

4. You argue in favor of rewards-based reinforcement learning, technically, even if these are metarewards reflecting the quality of reasoning as a whole. It appears from the current literature that rewards-based reasoning is not flexible/powerful enough for tasks you are discussing, and emergent concepts such as CURL (concave utility reinforcement learning) find their applications in the context of LMM. Are rewards crucial for your proposed framework? How CURL, for example, would fit instead of rewards?

**Discussion Potential:**

3

**Paper Summary:**

This position paper advocates applying metareasoning to Large Language Models to make models more robust in face of diverse tasks and limited external and internal knowledge.

**Position:**

Yes

**Position In Title:**

Yes

**Related Work:**

1

**Strengths And Weaknesses:**

Strength: The paper is well written and argues convincingly in favor of the need to extend the current mainstream approach to LLM with metareasoning capabilities. The challenges and proposed ways to address the challenges are clearly and formally stated. The proposed framework is described in sufficient detail to understand the principle but without too much burden on the reader, despite a broad of issues raised.

Weaknesses: it is my impression that the authors omit some important background and concurrent research, in particular rational metareasoning and bounded rationality, causality, and 'concave utility reinforcement learning'. It is also my impression that the novelty of this position is questionable, as other works have been recently presented integrating metareasoning with large language models, e.g. https://openreview.net/forum?id=ECXVwc1L4g .

See the questions section for more.

**Support:**

3

---

> ### Author Rebuttal · Authors · 2025-03-29
>
> Thanks for your thoughtful questions and bringing these interesting related papers!
>
> **Q1: Compared with other traditional meta-reasoning work - UAI 2012 paper**.
>
> Thanks for pointing us to the work “Selecting computations: theory and applications”.
> Our framework indeed aligns with the bi-level decision process, where a metalevel decision-making procedure governs the selection of object-level actions. This structure is central to many meta-reasoning approaches [1,2], including ours.
> However, a key distinction is that while the referenced work provides a theoretical foundation for selecting optimal metalevel decisions, our approach is data-driven, i,e., updating the metalevel decision—specifically, the reasoning strategy in our paper—through feedback obtained from multiple reasoning tasks.
>
> **Q2: Compared with recent meta-reasoning work - Rational meta-reasoning in LLMs**
>
> We have indeed cited this paper in the Self-Awareness module (Line 161). This work primarily focuses on adapting the thinking time based on the complexity of the input question by introducing a length regulariser to control the response length dynamically. They framed it as an online reinforcement learning (RL) problem, where the chain-of-thought (CoT) reasoning process is penalised based on token length, a technique also explored in other works such as DeepSeek R1.
>
> While this aligns with the self-awareness aspect of our framework—specifically, the ability to assess task difficulty and adjust reasoning depth (hence computational time)—it does not encompass the broader scope of our meta-reasoning framework. Our work extends beyond adaptive response length by incorporating multiple interacting components, including monitoring, evaluation & regulation, and meta-reflection. Most notably, we adopt a Bayesian meta-reasoning approach across multiple tasks, allowing for dynamic strategy adjustments informed by cumulative experience, which is not covered in the rational meta-reasoning paper.
>
> **Q3: What makes you think that pure Bayesian inference rather than causality calculus, is the right choice**
>
> Thank you for bringing causal reasoning into the discussion. We acknowledge that causal reasoning—especially counterfactual reasoning—can complement our framework, particularly in Section 4.2: Monitoring. Specifically, counterfactual reasoning could enable the model to explore “what-if” scenarios, verifying the correctness of its reasoning process in a counterfactual world without requiring access to ground truth. Such insights could serve as an additional form of meta-reward, enhancing the model’s ability to refine its reasoning strategies.
>
> Regarding our choice of Bayesian inference, we highlight the following advantages:
> - Bayesian inference provides a principled way to update beliefs, i.e., the initial reasoning strategy and LLMs’ prior belief.
> - LLM reasoning is inherently stochastic, particularly due to the probabilistic nature of sampling during response generation. - -Bayesian inference explicitly models this uncertainty, allowing for more robust decision-making under conditions of incomplete or noisy information.
>
> **Q4: Meta Rewards or concave utility reinforcement learning**
>
> We introduced meta-rewards to overcome the limitations of oversimplified reward structures in existing RL methods, making them more dynamic and multifaceted. Ideally, the reward in our framework is not predefined but can emerge from the iterative development of the LLM through self-play (see in Section 5: Self-play for meta-reward seeking). Instead of relying on a fixed reward model, our approach utilises a self-evolving evaluator LLM to provide adaptive feedback. The reward in our framework is thus just one format of feedback—adaptive and context-aware—rather than a static objective. So, in that sense, we agree that “*rewards-based reasoning is not flexible/powerful enough for tasks you are discussing*”.
>
> In terms of concave utility reinforcement learning (CURL), we recognise that it replaces conventional reward functions with a concave utility function, which is more flexible as it is derived from the agent’s behaviour distribution. This addresses critical issues such as reward hacking and over-simplification, which we highlighted in Section 2: Open Problem 3. While our framework does not explicitly adopt CURL, it aligns with the broader goal of mitigating the limitations of standard reward functions through adaptive, self-supervised signals.
>
> Please let us know if you have further concrete questions or concerns! Thanks for your engagement with our work!
>
> **References**
>
> [1] Meta-Cot: Generalizable Chain-Of-Thought Prompting In Mixed-Task Scenarios With Large Language Models.
>
> [2] Meta-Reasoning: Semantics-Symbol Deconstruction for Large Language Models

---

### Official Review · Reviewer_oVhh · 2025-03-14

**Significance:** 2
**Argument Clarity:** 2
**Rating:** 2
**Confidence:** 3

**Questions:**

- What are the benefits of the advocated framework over current reasoning models trained via RL such as DeepSeek R1?
- Could you produce a concrete example for the Meta-Reasoning framework, implementing both the inference and learning process?

**Discussion Potential:**

2

**Paper Summary:**

This paper advocates for a Bayesian Meta-Reasoning framework to make LLM reasoning more generalizable and robust.

In this framework, the LLM conducts Bayesian inference using the framework P(I, E, F | O), where I is the internal knowledge, E is the external view,  F is the reasoning strategy, and O is the observation.
The reasoning strategies and knowledge priors are updated during the learning process at the task and meta levels, respectively.

The authors also discuss the gaps between current studies and the framework, such as the challenge to accurately assess the self-awareness and the task solvability, insufficient adaptability to latent skill selection, and overlooked reasoning diversity.

Finally, the authors provide actionable insights such as better benchmarks for meta-reasoning evaluation and latent-space reasoning framework for LLMs.

**Position:**

Yes

**Position In Title:**

Yes

**Related Work:**

3

**Strengths And Weaknesses:**

Pro:
- Overall, the paper is well-written with illustrative figures for delivering the advocated framework.
- The meta-reasoning framework is intuitively correct and could be potentailly promising.

Cons:
- There are no proof-of-concept experiments to support the framework, despite some *actionable insights* being provided. This leads to my questions about the feasibility of the proposal and whether these additional complexities could provide extra benefits than pure RL.

- The discussions of alternative views are somewhat superficial, without questioning the REAL significance of the proposed meta-reasoning framework. For example, DeepSeek R1 does not predefine any reasoning or monitoring strategies while performing relatively well both in-domain (Code & Math) as well as in general tasks such as poem generation.

**Support:**

3

---

> ### Author Rebuttal · Authors · 2025-03-29
>
> Thanks for your valuable suggestions and questions!
>
> **Q1: No proof-of-concept experiments. Feasibility of the proposal, extra benefits than pure RL**
>
> As stated in the CFP, position papers are judged on the strength of their argument rather than experimental validation, such as [1,2] accepted by ICML 2024.
>
> Feasibility: Our proposed framework consists of multiple components, each of which has been examined through a thorough literature review. We have discussed existing approaches for realising these components, along with their limitations. This analysis already provides some indication of feasibility. Moreover, the learning process is based on Bayesian inference, which is a well-established and principled approach.
>
> Benefits:
> - self-awareness for adaptive reasoning. Unlike Deepseek-R1, which suffers from “overthinking”[3] and “forced thinking”[4], our framework dynamically adjusts reasoning strategies based on input complexity. This prevents unnecessary computation.
> - meta-learning capabilities. It naturally supports hierarchical probabilistic distributions over reasoning skills and dynamic update of reasoning strategies.
>
> **Q2:  Benefits over current reasoning models, especially generalisability**
>
> We acknowledge that pure RL-trained models have shown impressive generalisability. However, key limitations are observed:
> - Table 1 in [5] shows that Deepseek has inferior generalisation than o1-mini on math and scientific questions.
> - [6] shows that Deepseek can’t generalise well to role-playing tasks.
>
> Our framework could potentially further enhance the reasoning generalisability of pure RL (Section 2: open problems). Most RL-based approaches, like DeepSeek rely on static, oversimplified rewards, leading to reward hacking[7], i.e.,  achieve high reward scores without genuinely learning transferable reasoning patterns. Our two key solutions:
> - Meta-reward (Sec4.2 and *Self-play for meta-rewards seeking* ): an LLM evaluator identifies the reasoner LLM’s knowledge gap for a given task and adaptively provides feedback/reward to guide reasoning improvements iteratively. In that sense, the self-play mechanism eliminates the need for additional binary feedback from humans or a reward model [8].
> - Meta-thoughts (Sec4.3). Retrieves reasoning cases from past similar tasks to assist current reasoning. Instead of purely exploring and exploiting the decoding space as seen in existing inference scaling laws, the model learns from prior successful reasoning trajectories to make more informed steps. Preliminary findings [9,10] suggest that incorporating meta-thoughts significantly improves reasoning efficiency and accuracy.
>
> **Q3: A concrete example**
>
> Scientific hypothesis generation example.
>
> Initialisation:
> - I (Internal View): LLM’s parametric knowledge, e.g., general biomedical knowledge.
> - E (External View): Task-specific knowledge, such as retrieved relevant information from domain-specific knowledge bases， e.g., genomic databases.
>
> Bayesian Learning:
> - Updating reasoning strategies at the task-level.
> - Self-awareness to initiate the distribution of reasoning strategies, such as cross-domain analogy (identify analogous phenomena across fields), constraint satisfaction (formulate hypotheses that meet known constraints).
> - Choose an initial reasoning strategy F (e.g., cross-domain analogy)  informed by foundational and task-specific knowledge.
> - Generate reasoning process A using the current F, such as exploring Alzheimer’s disease using analogies from cancer research to generate novel hypotheses.
> - Follow A to generate hypotheses.
> - Evaluate the quality of hypotheses based on novelty, feasibility, plausibility,  and record feedback (O).
> - Update F using $p(F|O, I, E) \propto p(O|F, I, E)p(F|I, E)$
> - Updating knowledge priors at the meta-level.
> - Update knowledge priors I and E based on meta-rewards computed across multiple hypotheses generation tasks (e.g., Over multiple hypothesis-generation cycles in different neurodegenerative diseases, the model recognises a pattern that Single Nucleotide Polymorphism affecting microglial activation tend to be highly relevant. The LLM updates its priors to prioritise genes involved in neuroinflammation across future tasks).
>
> Bayesian Inference:
> - Generate hypotheses based on the chosen reasoning strategy
> - Evaluate the quality of hypotheses based on novelty, feasibility, plausibility, recorded as feedback O
>
> Please let us know if you have further concrete questions or concerns!
>
> **Ref**
>
> [1]Position: Tensor Networks are a Valuable Asset for Green AI
>
> [2]Position: Future Directions in the Theory of Graph Machine Learning
>
> [3]Do NOT Think That Much for 2+3=?
>
> [4]Output Length Effect on DeepSeek-R1’s Safety in Forced Thinking
>
> [5]ThinkBench
>
> [6]Reasoning Does Not Necessarily Improve Role-Playing Ability
>
> [7]Process Reinforcement Through Implicit Rewards
>
> [8]Self-Play Fine-Tuning Converts Weak Language Models to Strong Language Models
>
> [9]Buffer of thoughts
>
> [10]ReasonFlux

---

> > ### Comment · Reviewer_oVhh · 2025-04-02
> >
> > Thank you for your response. I greatly appreciate the value of thought experiments in research.
> >
> > - Regarding feasibility: The provided concrete example does not involve any actual inference or learning processes. I remain concerned about the proposal's feasibility (a concern also raised by reviewers CRfQ who noted it "might not be realistic" and 1XTN who called it "only a tentative plan"). While I understand that experiments are not necessary for a position paper, I am still requesting a proof-of-concept example in a real-world context (e.g., prompting ChatGPT models) to demonstrate the potential of the proposed approach. Ultimately, these ideas would need to be implemented in real executable systems based on LLMs.
> >
> > - Regarding Q2: The key limitations you mentioned on mathematical reasoning and role-playing are not valid comparisons. R1 and O-series do not use the same base model, and role-playing capabilities could be easily enhanced by incorporating relevant corpus data. Therefore, these "limitations" should not be attributed to pure RL constraints. The mentioned ReasonFlux adopts Gemini 2.0 to curate templates highly relevant to the mathematical problems, and they ONLY evaluate on MATH/AIME datasets, which cannot support your claim that this could enhance reasoning generalizability, as  (i) the trained model and the templates only suitable for math while fail for coding tasks; (ii) the inclusion of external Gemini models might introduce potential leak and noise.
> >
> > Given that my concerns remain unaddressed, I maintain my original rating.

---

> > > ### Author Response · Authors · 2025-04-04
> > >
> > > Q1:  As requested, we prompt the ChatGPT to show how this system works in a simplified setup with human intervention.
> > >
> > > **Input question**
> > > > A tutor program, one for math, another for english, both aim to help students pass an exam. Based on the data, do you think the tutorial is effective for student improvement? 60% of students in the program passed at least one project, while 45% of students not in the program passed at least one exam.
> > >
> > > **Direct Answer from ChatGPT**
> > > > This suggests that the tutoring program may indeed **have a positive impact** on student improvement, especially since the program seems to help a higher proportion of students succeed in at least one project
> > >
> > > **Our framework** (Results denotes generated response from ChatGPT; distinguishes for each module compared with existing RL are in red )
> > >
> > > **Self-awareness** ``` Before directly answering the question, assess the model's knowledge gap and enable them to retrieve necessary knowledge. can be done using Bayesian learning for uncertainty measurement, prior distribution estimation for required skills. ```
> > > > prompt: Adding “Before answering the question, consider what knowledge is necessary to enhance the quality of your response... you've incorporated this knowledge...” to original input question
> > >
> > > > Results:  *Essential **Statistical Concepts**: To evaluate effectiveness, we should consider: Causal Inference vs. Correlation: ...*
> > >
> > > > Effects: LLM realises its knowledge gap and incorporates statistical concepts as conditional input.
> > >
> > > **Monitoring**
> > > ``` (i) Reward/feedback is provided by a self-evolving LLMs, beyond predefined and oversimplified; (ii) internal explicit thoughts into latent space to address superficial token-level match```
> > >
> > > > Prompt: Not applicable, as we can’t manipulate the internal generation process in prompting.
> > >
> > > **Evaluate and regulation** ```Enable feedback from similar cases, external tools, etc```
> > >
> > > > Prompt: we manually provide feedback  "... This situation might be an instance of Simpson's paradox—where trends that appear in separate groups reverse when the groups are combined.."
> > >
> > > > Results: Yes, ... this is a classic case where Simpson's Paradox ... the **overall trend can sometimes mask or reverse the trends** seen within individual groups ...
> > >
> > > **Meta-reflection (learning)** ```A bi-level update: model parameter and the prior of reasoning strategy based on required skills, for example,  we can use modular training for different skills based on the prior```
> > > > Prompt:  Can you summarise the lesson you have learnt from this case and think about how to apply it to other reasoning cases?
> > >
> > > > Results: *The key takeaway ... is careful data interpretation ... statistical paradoxes ...*”
> > >
> > > > Effects: Beyond correct the previous case, LLM also updates its prior distribution of required skills, i.e., by incorporating statistical paradoxes for general causal reasoning.
> > >
> > > **Inference other similar cases**
> > >
> > > > Prompt: Add the *Results* in meta-reflection as an additional context for another reasoning case in **medical domain**: *A treatment is applied to two groups, young and old,  the overall recovery rate is 65% in the treatment group, overall recovery rate is 60% in the controlled group, do you think  the treatment is effective?*
> > >
> > > > Results: *... it might also be an instance of Simpson's Paradox—the overall recovery rates ...**may not fully reflect the effectiveness of the treatment**  … **What we need to do is Recovery Rates within Each Age Group**…*
> > >
> > > > Effects: LLM leverages the updated belief, i.e., paradox to correctly proceed the problem in the new (medical) domain, e.g., by requesting data with each group.```
> > >
> > > Q2:
> > >
> > > (1) [5,6] are examples, showing that there is room for DeepSeek-R1 to be enhanced in generalisation, similar to effect of you mentioned example in poem to show its generalisability. We apologise for the term “*key limitations*” for any caused misunderstanding, should be “undesirable generalisation cases are still observed”.
> > >
> > > (2) One of the key limitations in existing RL is the reward hacking problem, and the actionable insights are self-play paradigm. Related discussion in Question 4 in Review from 8n3F and our response in Q4. Other key limitations/improvements please refer to the text in red in Q1.
> > >
> > > (3) Citing ReasonFlux is to show by incorporating the shared reasoning pattern (the Simpson paradox example in Q1), i.e., the template for different samples in ReasonFlux can enhance generalisability. The generalisability can be OOD, also can be across a range of similar problems, rather than being limited to specific instances, which is the highlight in ReasonFlux.
> > >
> > > Our framework can also enhance the OOD, as illustrated in Q1, one in education, another in medicine. Here, the generasability is achieved via sharing common reasoning pattern, the Simpson's paradox.
> > >
> > > Please let us know if you have further concrete questions or concerns that we can address. Thank you for your engagement!

---

### Decision · Program_Chairs · 2025-04-30

**Decision:**

Accept (poster)

**Comment:**

Important area for research.   Thought provoking paper.
The discussions of alternative views are somewhat superficial.  For instance, more discussion with recent reasoning using RL methods should be done, which the authors discuss, but I believe more is needed in the paper, given the extensive recent research here..
One reviewer was strongly in support of Weak Reject, motivated by their wanting to see more rational meta-reasoning discussed.
Extensive discussion shows engagement.
One criticism is lack of evidence in support and thus feasibility.  The very specific nature of the proposal leads to questions, but the authors include a "Gaps and Questions" section.